# Construction of 5-(Alkylamino)-6-aryl/alkylpyrazine-2,3-dicarbonitriles via the One-Pot Reaction of Alkyl Isocyanides with Aryl/Alkyl Carbonyl Chlorides and Diaminomaleonitrile: Fluorescence and Antimicrobial Activity Evaluation

**DOI:** 10.3390/molecules27238278

**Published:** 2022-11-27

**Authors:** Amal Al-Azmi, Elizabeth John

**Affiliations:** Chemistry Department, Kuwait University, P.O. Box 5969, Safat 13060, Kuwait

**Keywords:** pyrazine, isocyanide, diaminomaleonitrile, fluorescence, antimicrobial

## Abstract

5-(Alkylamino)-6-aryl/alkylpyrazine-2,3-dicarbonitriles were successfully synthesized in good to moderate yields by reacting alkyl isocyanides with aryl/alkyl carbonyl chlorides, followed by the addition of diaminomaleonitrile. The synthesized pyrazines were fully characterized in this investigation, and X-ray crystal structure analysis was performed on some derivatives. The antibacterial and antifungal activities of the newly synthesized pyrazine-2,3-dicarbonitriles were assessed in addition to their UV and fluorescence results. All the compounds showed similar UV–Vis spectral features with absorption peaks (λ_max_) around 267, 303, and 373 nm.

## 1. Introduction

Researchers worldwide have been eagerly designing new synthetic routes to many *N*-heterocyclic molecules for decades due to their highly notable role as bioactive materials [1,2,3,4,5,6,7,8,9,10,11,12,13]. Among these heterocycles are pyrazines. Pyrazines **1** are well-known nitrogen-containing heterocyclic compounds with two nitrogen atoms in 1,4-positions of the six-membered ring [4,5,6,7,8,9,10] (Figure 1).

They are considered one of the most significant classes of nitrogen heterocyclic compounds, and they are either obtained naturally [6,7,11] or synthesized chemically [1,2,3,4,5,6,7,9,10,11,12,13,14,15]. Pyrazines are found naturally in many living organisms, including animals, plants, and marine organisms, such as pyrazines **2** and **3**, and insects [5] (Figure 1).

Generally, pyrazines exhibit vital different biological actions and are recognized to be fundamental in numerous approved drugs [1,2,3,4,5,6,7,8,9,10,11,12,13]. A pyrazine moiety is found in riboflavin, known as vitamin B2 and folic acid; in addition, they are abundant in many well-known pharmaceuticals, such as anticancer and [14] antibacterial agents. For example, pyrazinamide **4** is the most common synthetic antibacterial agent used in the treatment of tuberculosis, whereas sulfametopyrazine **5** is used in treating UTIs, chronic bronchitis, diabetes, and tuberculosis and as a diuretic [6,7,14,15,16,17,18,19] (Figure 2).

Furthermore, pyrazines play a key role in the fragrance industry; alkyl pyrazines, for instance, are a vital element due to their strong olfactory effects [20]. In addition, a group of low-bandgap π-conjugated pyrazine polymers is utilized in photovoltaic devices, often known as solar cells [21,22].

There are a wide variety of active substituted pyrazine derivatives. These derivatives include pyrazine-2,3-dicarbonitriles **7**, which are produced from the condensation of diaminomaleonitrile (DAMN) **6** [1,2,3] with α-diketones (Figure 3). They were extensively investigated and discovered to be important in food, agricultural, dyes, pigments, and pharmaceutical areas because of their interesting chemical properties [1,2,3,4,5,23].

Recently, several substituted dicyanopyrazines have been described as electroluminescent reagents and synthetic coloring substances. A comparison between phthalonitrile compounds and dicyanopyrazines was conducted, and it was found that their derivatives exhibited significant features such as strong fluorescence, intense chromophoric effects, and high solubility in polar solvents in addition to a high freezing point. Such features emerged from the two aza functions in the 1,4-position of the corresponding pyrazine ring [24]. The dinitrile derivative can readily react with many nucleophiles, such as primary and secondary amines, alcoholates, enamines, and thiolates, to produce various substituted pyrazines.

The genesis of isocyanides reaction with acyl chlorides was discovered by Nef. The reaction produces imidoyl chlorides, which can be hydrolyzed or trapped to furnish diverse cyclic products [25]. In contrast to the Ugi and Passerini reactions, the reaction between isocyanides and acyl chlorides usually necessitates heating to yield acceptable moderate yields, as illustrated in Figure 1 [26].

Shaabani et al. reported synthesizing a group of 1,6-dihydropyrazine-2,3-dicarbonitriles obtained from multicomponent reactions (MCRs) of DAMN **6**, ketones, and isocyanides. The synthesized pyrazines were subsequentially mixed with different isocyanates or isothiocyanates derivatives that produced novel highly substituted imidazo[1,5-*a*] pyrazines [27].

For many years, our group has explored different routes for developing new types of nitrogen-heterocyclic compounds derived from DAMN **6**, which is a rich source of multi-nitrogen-containing products with numerous applications mainly in bioactive [14,15,16,28,29,30] or OLED materials [31] with the increased demand for new highly substituted pyrazine-2,3-dicarbonitriles. Herein, we report the synthesis of a new category of pyrazine-2,3-dicarbonitrile derivatives and the outcome of their antimicrobial and fluorescence activities.

## 2. Results and Discussion

The reactions of isocyanides **12** with various carbonyl compounds **9**, and DAMN **6** are well-known in the literature [1,2,3]. However, their reaction with substituted carbonyl chlorides and DAMN **6** has not been scrutinized. Our investigation began by adding an equimolar amount of cyclohexyl isocyanide **12** to benzoyl chloride **9** and heating the mixture for 1 h at 60 °C. The reaction mixture color changed from colorless to light brown, a solution of DAMN **6** (1.0 mmol) in dry tetrahydrofuran (10 mL) was added to the reaction mixture, a precipitate started to evolve, and the stirring was continued for 12 h at room temperature. The light brown precipitate was then filtered off and washed with petroleum ether (40–60 °C). Surprisingly, while adding petroleum ether (40–60 °C) to the filtrate, a yellow precipitate formed, which was filtered and further purified by being washed with petroleum ether (40–60 °C) several times to yield the pure product **13a**. The spectroscopic analyses of the yellow precipitate, as revealed by ^1^H NMR spectra, indicated the presence of multiplets at δ = 1.19–1.28, 1.41–1.51, 1.66–1.77, 2.00–2.04, and δ = 3.99–4.05 for the cyclohexyl ring protons, doublet at δ = 5.77, *J* = 7.6 Hz to the NH protons and multiplets at δ = 7.59–7.67 Hz and δ = 7.63–7.61 Hz for the phenyl protons. The ^1^H decoupled ^13^C-NMR spectrum of **13a** showed cyclohexyl carbons resonating at 24.7, 25.5, 32.4, and 50.6 ppm, respectively. The X-ray confirmed that the isolated yellow precipitate was pyrazine-2,3-dicarbonitrile **13a** (Figure 4).

A group of pyrazines **13b**–**f** was prepared (Table 1). The NMR spectrum of pyrazine **13f** exhibited four multiplets at δ = 1.22–2.07 and δ = 4.03 due to the cyclohexyl ring protons, doublet at δ = 5.81, *J* = 7.6 Hz due to the NH protons, and triplets at δ = 7.45 and δ = 7.52, doublet at δ = 7.66 and multiplets at δ = 7.75–7.83 Hz for aromatic hydrogens. The ^1^H decoupled ^13^C-NMR spectrum of **13f** showed 17 distinct resonances in agreement with the suggested structure, with the cyclohexyl carbons resonating at 24.8, 25.6, 32.5, and 50.7 ppm, respectively. The mass spectra showed a peak at 379, and HRMS 379.1790 agreed with the mass calculated for the product formulae C_24_H_21_N_5_, which was 379.1791. The structure of compound **13f** was finally unambiguously confirmed by X-ray crystallographic analysis (Figure 4).

On the other hand, when *t*-butyl isocyanide **12** was used 5-(*t*-butylamino)-6-arylpyrazine-2,3-dicarbonitriles **13h-k** were obtained in low yields of 12–29% under the conditions described above. Hence, the yield improved when the temperature was lowered to 25–30 °C, and the time was reduced to 15–30 min at 37–43% (Figure 2). The brown solid precipitated out in relatively small amounts from the reaction mixture. Spectroscopic analyses showed that the solid was amide **14a**. This finding indicates that during the reaction of isocyanide with carbonyl chloride derivatives, traces of the latter did not participate in the formation of the targeted intermediate imidoyl chloride **15** and consequently reacted with DAMN **6** to form amide **14** (Figure 3).

The mechanism of pyrazines **13** formation is shown in Figure 2. The initially formed intermediate is the imidoyl chloride **15**, which resulted from the attack of alkyl isocyanide on the electrophilic carbonyl group of the carbonyl chloride. This highly reactive electrophilic intermediate was then attacked by DAMN **6**, which underwent additions followed by HCl elimination. Eventually, intramolecular cyclization was followed by the loss of H_2_O molecules, which led to the final product **13**.

There have been many attempts to grow single crystals of the newly synthesized compounds; good-quality **13a**, **e**, **f**, and **k** crystals have been obtained. The crystal structures of these compounds are shown in Figure 4. The molecular structure information of these compounds obtained from the single-crystal X-ray diffraction method perfectly agrees with the predicted synthetic protocol and other characterization techniques, such as NMR and mass spectroscopy. The structure of these compounds in crystal state is consistent with non-planar geometry, with the cyclohexyl groups oriented orthogonally against the pyrazine ring and the phenyl rings twisted around 40° for pyrazine. In the case of compounds containing biphenyl fragments (**13f** and **13k**), both phenyl groups in biphenyl are oriented in different planes, with a torsion angle of around 30°. This orientation causes the pyrazine rings and the second phenyl fragment of the biphenyl group to achieve co-planar orientation. We observed that all these compounds in their crystal structure exhibited appreciable intermolecular π–π interactions (Figure 5). In addition, every compound showed intermolecular and/or intramolecular H-bonding interactions in its crystal network (Figure 6). These π–π interactions, as well as H-bonding interactions and particular molecular geometry, are expected to significantly influence their electronic and fluorescence characteristics. We also expected that these non-bonding interactions could sustain their dissolved solution state. A detailed discussion of the crystal data of these molecules is provided in the Appendix A.

Crystallographic data for the structure of compounds **13a**, **e**, **f**, and **k** reported in this paper were deposited at the Cambridge Crystallographic Data Centre (CCDC) as a supplementary publication, with CCDC numbers 2213188, 2213187, 2213186, and 2213189, respectively. See Appendix A.

The scope of this reaction can be extended to prepare many pyrazines with different substituents for both C-5 and C-6 by choosing the correct substituted carbonyl chloride and isocyanide derivatives, respectively. This strategy is shown in pyrazines **13f** and **13g**. Generally, this method is convenient for many pyrazine-2,3-dicarbonitrile analogs that can be potentially useful in applications, especially in the OLED and pharmaceutical industries.

## 3. Fluorescence

We investigated the fluorescent characteristics of samples **13a** to **13k**. In the present study, our goal was to gain insight into the effects of phenyl/biphenyl/pyridyl and cyclohexyl substituents on the fluorescence properties of dicyanopyrazine moiety. All the tested compounds showed similar UV–Vis spectral features with absorption peaks (λ_max_) around 267, 303, and 373 nm. As a result, we monitored the fluorescence characteristics of these compounds by exciting the molecules in the above wavelengths and recording their emission spectra. The experimental setup was such that all the chromophores were dissolved in freshly distilled DCM at a 1 × 10^−6^ molar concentration. All the emission spectra were recorded on the same day at room temperature (25 °C).

All 11 samples (**13a** to **13k**) showed appreciable fluorescence emission at the three tested excitation wavelengths of 267, 303, and 373 nm, and their emission characteristics depend on the substituents present in their structure. We observed that chromophores containing both phenyl/4-substituted phenyl and cyclohexyl rings (**13a**, **13b**, **13c**, **13d**, and **13f**) showed enhanced emission (in the range of 2–8 × 10^6^ CPS) at around the 450–460 nm region when excited at 373 nm. Furthermore, their fluorescence intensity is intermediate upon the 303 nm excitation (in the range of 1–4 × 10^6^ CPS) and is the least at the 267 nm excitation (in the range of 2–6 × 10^5^ CPS). However, if phenyl/4-substituted phenyl rings are absent in the structure (**13e**, **13g**, and **13h**), a hypochromic shift in the fluorescent spectra (to around 400–420 nm) was observed, and the 303 nm excitation provided more intense emissions than the 373 nm excitation. The absence of cyclohexyl rings (**13h**, **13i**, and **13j**) decreased the fluorescent intensity except for **13k** (possessing almost the same fluorescence characteristics as **13b**), which contains a biphenyl moiety on the pyrazine ring.

When comparing the effects of phenyl substitution upon the phenyl-, cyclohexyl-substituted pyrazine systems, we found that **13c** exhibited good fluorescence (8 × 10^5^) at 454 nm and was almost double the intensity of **13a** (4 × 10^5^) upon excitation at 267 nm (Figure 7). At the same excitation wavelength, **13b** exhibited almost the same fluorescence intensity as **13a** and both **13d** and **13f** analogs, which showed diminished fluorescent intensity (2.2 × 10^5^) compared with **13a**. In addition, **13b** exhibited enhanced fluorescence emission at 314 and 611 nm compared with the other analogs, which all have minimal fluorescence at the 267 nm excitation (Figure 7). At 303 and 373 nm excitation, all four tested samples showed the same fluorescence features with intense emission around 455 nm, as mentioned above. In the case of the 303 nm excitation, the trend in fluorescence intensity was **13c** > **13a** > **13b** > **13d** ≈ **13f**. For 373 nm, the order was **13c** > **13b** >**13a**> **13d** ≈ **13f**, as shown in Table 2. Notably, the enhanced fluorescence intensity of these chromophores (almost double) upon chloro-substitution at the phenyl ring and the decreased fluorescence (almost half) for methoxy/biphenyl substituents compared with unsubstituted phenyl derivatives. At the same time, pyrazine **13k** is a superior fluorescent emitter to phenyl: *t*-butyl substituted analog **13i**. These trends are interesting since the fluorescence properties are directly linked to variations in the electronic states of these chromophores upon substitution. The increased fluorescence always has an added significance in that this trend allows employing such derivatives as useful florescent markers for biological or commercial systems. The quantitative and theoretical aspects of the fluorescent characteristics of these compounds are currently being investigated.

## 4. Pharmacology

### 4.1. Methodology

The antimicrobial activities of 11 pyrazine derivatives **13a**–**k** were tested using the Agar-well diffusion technique against six different microbial cultures. Pure cultures of *Escherichia coli* and *Pseudomonas aeruginosa* (Gram-negative bacteria), *Bacillus subtilis* and *Staphylococcus aureus* (Gram-positive bacteria), and *Candida albicans* and *Saccharomyces cerevisiae* (yeast) were involved in the test. An aliquot of 0.1 mL of each bacterial strain was inoculated and spread onto nutrient agar (NA), while 0.1 mL of the yeast was spread onto potato dextrose agar (PDA). The inoculated plates were supplied with 70 µL of each tested chemical (dissolved in DMSO solvent) with a total final concentration of 1mg mL^−1^. The chemicals were included in 7 mm wells produced by a sterile cork borer. The NA plates were incubated at 37 °C for 24 h, whereas PDA plates were incubated at 30 °C for 48 h. We determined and averaged the inhibition zones around the wells based on three recorded replications. In this experiment, 1 mg/mL^−1^ samples of Cycloheximide, Penicillin G, Kanamycin, and Ampicillin were used as references. Cycloheximide inhibits eukaryotic organisms, whereas Penicillin G, Kanamycin, and Ampicillin inhibit prokaryotes.

Some of the tested chemicals in their dissolved form showed inhibition against some tested microbes. The negative control (DMSO as a solvent) also showed no inhibition zone. The positive controls showed inhibition zones. The values are listed in Table 3.

According to Table 3, inhibition zone <4 present no inhibition. Inhibition zone 5–9 mm is weak. The inhibition zone between 10 and 20 mm is intermediate, and >20 is strong. Therefore, most chemical compounds investigated in the current study showed no or weak inhibition against Gram-positive and Gram-negative bacteria and the tested yeast strains. Four chemical compounds, i.e., **13a**, **d**, **e**, **f**, showed an intermediate effect on the Gram-negative bacteria *Pseudomonas aeruginosa*. Chemicals **13b** and **13f** have a weak effect on *Candida albicans*. The prepared chemicals performed mild inhibition when comparing the inhibition effect of 1 mg mL^−1^ of the prepared chemical compounds on Gram-positive and negative bacteria and the effect of 1 mg mL^−1^ of ampicillin. The same is correct when comparing the cycloheximide effect on *Candida albican*s with the effect of the tested chemicals. Table 4.

### 4.2. Experimental Section

#### 4.2.1. General

All the reactions were conducted under a nitrogen atmosphere unless otherwise noted. All the analyses were conducted in the Research Sector Projects Unit (RSPU) at Kuwait. TLC was performed using Polygram sil G/UV 254 TLC plates, and visualization was conducted by ultraviolet lights at 254 and 350 nm. Column chromatography was performed using Merck silica gel 60 of mesh size 0.040–0.063 mm. ^1^H and ^13^C NMR spectra were recorded using Bruker NEO 400 MHz and Bruker Avance II 600 MHz superconducting NMR spectrometers. Mass spectra were recorded with a GCMS-DFS-Thermo, high-resolution spectrometer. Fluorescence measurements were conducted with a Jobin Yvon—Fluoromax-4, Spectrofluorometer using a 1 cm path length cuvette at room temperature. The single-crystal X-ray analysis was conducted on Rigaku Rapid II and Bruker X8 prospector diffractometers. Melting points were determined via differential scanning calorimetry (DSC) analyses on Shimadzu DSC-50.

#### 4.2.2. General Procedure for the Synthesis of Compounds **13a**–**k**, **14c**, **j**, **k**

In a typical experimental procedure, a dry, two-necked, 50 mL round-bottom flask was charged with 1.0 mmol of substituted carbonyl chloride derivatives and 1.0 mmol of cyclohexyl isocyanide, heated at 60 °C for 1 h with *t-*butyl isocyanide, and heated for 25–30 min at 25–30 °C. Dry tetrahydrofuran (10 mL) and DAMN (1.0 mmol) were added to the reaction mixture and stirred at room temperature for 12 h. The solid was filtered and crystallized to produce compounds **14**. Petroleum ether was added to the filtrate to furnish compounds **13a**–**k**.

*5-(Cyclohexylamino)-6-phenylpyrazine-2,3-dicarbonitrile* **13a**. Yellow crystals crystallized from EtOH (88%), mp 120 °C; ν_max_ (KBr)/cm^−1^: 3411, 2939, 2852, 2227, 1579, 1561, 1525, 1502, 1449, 1392, 1367, 1331, 1278, 1092, 1081, 1011, 764, 713; ^1^H NMR 400 (CDCl_3_): δ 1.19–1.28 (m, 3H), 1.41–1.51 (m, 2H), 1.66–1.77 (m, 3H), 2.00–2.04 (m, 2H), 3.99–4.05 (m, 1H), 5.77 (d, 1H, *J* = 7.6 Hz), 7.59–7.61 (m, 1H), 7.63–7.67 (m, 1H); ^13^C NMR 600 (CDCl_3_): δ 24.7, 25.5, 32.4, 50.6, 114.2, 114.8, 119.7, 128.1, 130.0, 131.3, 131.4, 133.8, 145.5, 151.3, *m*/*z* (EI): 303 (M+); *m*/*z* (EI): 303.1478 (M+, C_18_H_17_N_5_ calcd, 303.1478).

*5-(Cyclohexylamino)-6-p-tolylphenylpyrazine-2,3-dicarbonitrile***13b**. Yellow crystals crystallized from EtOH (78%), mp 157–158 °C; ν_max_ (KBr)/cm^−1^: 3412, 2932, 2856, 2227, 1558, 1528, 1504, 1451, 1396, 1366, 1275, 1186, 1084, 731; ^1^H NMR 400 (CDCl_3_): δ 1.18–1.29 (m, 3H), 1.42–1.52 (m, 2H), 1.59 (s, NH), 1.66–1.77 (m, 3H), 2.00–2.04 (m, 2H), 2.48 (s, 3H), 3.96–4.05 (m, 1H), 5.76 (d, 1H, *J* = 7.6 Hz), 7.39 (d, 2H, *J* = 8.0 Hz), 7.56 (d, 2H, *J* = 8.0 Hz); ^13^C NMR 600 (CDCl_3_): δ 21.7, 24.7, 25.6, 32.4, 50.6, 53.6, 114.3, 114.9, 119.8, 128.0, 130.7, 130.98, 131.0, 142.0, 145.6, 151.3; *m*/*z* (EI): 317(M+); *m*/*z* (EI): 319.1792 (M+, C_19_H_21_N_5_ calcd, 319.1791).

*5-(4-Chlorophenyl)-6-(cyclohexylamino)pyrazine-2,3-dicarbonitrile* **13c**. Yellow crystals crystallized from EtOH (83%), mp 129–130 °C; ν_max_ (KBr)/cm^−1^: 3407, 3388, 2933, 2856, 2226, 1666, 1593, 1571, 1558, 1527, 1451, 1396, 1366, 1278, 1093, 1008, 913, 844, 742; ^1^H NMR 400 (CDCl_3_): δ 1.19–1.31 (m, 3H), 1.42–1.52 (m, 3H), 1.67–1.78 (m, 3H), 2.00–2.04 (m, 3H), 3.99–4.05 (m, 1H), 5.67 (d, 1H, *J* = 7.6 Hz), 7.57 (d, 2H, *J* = 8.4 Hz), 7.63 (d, 2H, *J* = 8.4 Hz); ^13^C NMR 600 (CDCl_3_): δ 24.7, 25.5, 32.4, 50.8, 114.0, 114.6, 119.7, 129.6, 130.3, 131.4, 131.4, 132.2, 137.7, 144.2, 151.2; *m*/*z* (EI): 339(M+); *m*/*z* (EI): 339.1247 (M+, C_18_H_17_N_5_Cl calcd, 339.1245).

*5-(Cyclohexylamino)-6-(4-methoxyphenyl)pyrazine-2,3-dicarbonitrile* **13d**. Yellow crystals crystallized from EtOH (80%), mp 163–164 °C; ν_max_ (KBr)/cm^−1^: 3388, 2933, 2856, 2227, 1666, 1593, 1558, 1528, 1504, 1450, 1422, 1396, 1366, 1278, 1092, 1009, 908, 843, 733; ^1^H NMR 400 (CDCl_3_): δ 1.19–1.28 (m, 3H), 1.41–1.51 (m, 2H), 1.66–1.77 (m, 3H), 2.00–2.04 (m, 2H), 3.99–4.05 (m, 1H), 5.77 (d, 1H, *J* = 7.6 Hz), 7.59–7.61 (m, 1H), 7.63–7.67 (m, 1H); ^13^C NMR 600 (CDCl_3_): δ 24.7, 25.6, 32.5, 50.6, 55.8, 114.3, 114.9, 115.4, 119.8, 126.0, 129.8, 130.6, 145.3, 151.2, 162.1, *m*/*z* (EI): 335(M+); *m*/*z* (EI): 335.1743 (M+, C_19_H_21_N_5_O calcd, 335.1741).

*5-(Cyclohexylamino)-6-methylpyrazine-2,3-dicarbonitrile* **13e**. Yellow crystals crystallized from pet. ether (72%), mp 190–191 °C; ν_max_ (KBr)/cm^−1^: 3378, 2941, 2923, 2856, 2245, 2219, 1582, 1514, 1408, 1367, 1257, 1077, 971, 604; ^1^H NMR 600 (CDCl_3_): δ 1.26–1.32 (m, 3H), 1.45–1.51 (m, 2H), 1.64 (s, 2H), 1.72–1.75 (m, 1H), 1.80–1.84 (m, 2H), 2.06–2.09 (m, 2H), 2.47 (s, 3H), 4.01–4.04 (m, 1H), 5.09 (d, 1H, *J* = 6.0 Hz); ^13^C NMR 600 (CDCl_3_): δ 15.6, 19.9, 20.7, 27.8, 45.7, 109.3, 109.9, 114.3, 126.1, 140.1, 147.0; *m*/*z* (EI): 241(M+); *m*/*z* (EI): 241.1323 (M+, C_13_H_15_N_5_ calcd, 241.1322).

*5-(Cyclohexylamino)-6-biphenylpyrazine-2,3-dicarbonitrile* **13f**. Yellow crystals crystallized from pet. ether (40%), mp 123 °C; ν_max_ (KBr)/cm^−1^: 3413, 3359, 3932, 2855, 2227, 1563, 1503, 1396, 1276, 1061, 1007, 909, 735; ^1^H NMR 400 (CDCl_3_): δ 1.22–1.31 (m, 2H), 1.44–1.53 (m, 2H), 1.68–1.80 (m, 4H), 2.03–2.07 (m, 2H), 4.03–4.06 (m, 1H), 5.81 (d, 1H, *J* = 7.6 Hz), 7.45 (tt, 1H, *J* = 7.3, 2.1), 7.52 (t, 2H, *J* = 7.7), 7.66 (d, 2H, *J* = 7.2), 7.75–7.83 (m, 4H); ^13^C NMR 600 (CDCl_3_): δ 24.8, 25.6, 32.5, 50.7, 114.3, 114.8, 119.8, 127.4, 128.6, 128.6, 129.3, 131.3, 132.6, 139.7, 144.4, 145.1, 151.3; *m*/*z* (EI): 379 (M+); *m*/*z* (EI): 379.1790 (M+, C_24_H_21_N_5_ calcd, 379.1791).

*5-(Cyclohexylamino)-6-(5-(trifluoromethyl)pyridine-2-yl)pyrazine-2,3-dicarbonitrile* **13g**. Yellow crystals purified by column chromatography using pet. ether: DCM: ethyl acetate 4:2:1 (87%), mp 156–157 °C; ν_max_ (KBr)/cm^−1^: 3328, 3284, 3203, 2934, 2857, 2224, 1607, 1585, 1525, 1501, 1448, 1327, 1304, 1170, 1140, 1082, 1019, 858, 732; ^1^H NMR 400 (CDCl_3_): δ 1.38–1.59 (m, 5H), 1.68–1.75 (m, 1H), 1.79–1.82 (m, 2H), 2.04–2.08 (m, 2H), 4.16–4.22 (m, 1H), 8.18–8.21 (dd, 1H, *J* = 8.8, 2.0), 8.79 (d, 1H, *J* = 8.4), 8.97 (s, 1H), 10.75 (s, 1H); ^13^C NMR 600 (CDCl_3_): δ 24.5, 25.8, 32.3, 50.1, 113.9, 114.8, 117.2, 124.1, 127.5, 127.7, 133.3, 134.3, 134.82, 134.85, 144.4, 152.1, 157.3; *m*/*z* (EI): 372(M+); *m*/*z* (EI): 372.1305 (M+, C_18_H_15_F_3_N_6_ calcd, 372.1305).

*5-(t-Butylamino)-6-methylpyrazine-2,3-dicarbonitrile* **13h**. Pale yellow crystals purified by column chromatography using pet. ether: DCM: ethyl acetate 4:2:1 (37%), mp 139–140 °C; ν_max_ (KBr)/cm^−1^: 3401, 2973, 2918, 2229, 1567, 1514, 1455, 1399, 1368, 1242, 1211, 911, 734; ^1^H NMR 400 (CDCl_3_): δ 1.53 (s, 9H), 2.45 (s, 3H), 5.09 (s, 1H); ^13^C NMR 600 (CDCl_3_): δ 20.8, 28.6, 54.0, 114.3, 114.9, 119.1, 130.3, 145.7, 152.3; *m*/*z* (EI): 215(M+); *m*/*z* (EI): 215.1165 (M+, C_11_H_13_N_5_ calcd, 215.1165).

*5-(t-Butylamino)-6-phenylpyrazine-2,3-dicarbonitrile* **13i**. Pale yellow crystals purified by column chromatography using pet. ether: DCM: ethyl acetate 4:2:1 (39%), mp 167–168 °C; ν_max_ (KBr)/cm^−1^: 3290, 2972, 2231, 1647, 1578, 1534, 1502, 1448, 1434, 1395, 1367, 1200, 1181, 944, 912, 731, 691; ^1^H NMR 400 (CDCl_3_): δ 1.58 (s, 9H), 7.53 (t, 2H, *J* = 8.0 Hz), 7.67 (t, 1H, *J* = 8.4 Hz), 7.95 (d, 2H, *J* = 7.6 Hz), 9.45 (s, 1H); ^13^C NMR 600 (CDCl_3_): δ 28.5, 54.3, 113.3, 114.0, 116.5, 128.6, 131.3, 132.3, 133.9, 134.4, 136.1, 153.1; *m*/*z* (EI): 277(M+); *m*/*z* (EI): 277.1323 (M+, C_16_H_15_N_5_ calcd, 277.1322).

*5-(t-Butylamino)-6-(4-methoxyphenyl)pyrazine-2,3-dicarbonitrile* **13j**. Pale yellow crystals purified by column chromatography using pet. ether: DCM: ethyl acetate 4:2:1 (41%), mp 239–240 °C; ν_max_ (KBr)/cm^−1^: 3416, 3327, 3218, 2978, 2893, 2802, 2709, 2600, 2501, 2208, 2080, 1643, 1608, 1523, 1502, 1402, 1380, 1322, 1304, 1250, 1179, 1031, 844, 760; ^1^H NMR 400 (DMSO): δ 1.26 (s, 9H), 3.83 (s, 3H), 7.04 (d, 2H, *J* = 8.4 Hz), 7.40 (s, 2H), 7.93 (d, 1H, *J* = 8.8 Hz), 9.56 (s, 1H); ^13^C NMR 600 (DMSO): δ 27.1, 50.9, 55.4, 89.9, 113.5, 113.9, 116.9, 124.9, 126.8, 129.9, 162.2, 164.6; *m*/*z* (EI): 307(M+); *m*/*z* (EI): 307.1425 (M+, C_17_H_17_ON_5_ calcd, 307.1428).

*5-(t-Butylamino)-6-biphenylpyrazine-2,3-dicarbonitrile* **13k**. Yellow crystals crystallized from pet. ether (43%), mp 189 °C; ν_max_ (KBr)/cm^−1^: 3418, 2971, 2929, 2228, 1683, 1607, 1562, 1504, 1454, 1395, 1367, 1274, 1207, 1007, 851, 765; ^1^H NMR 400 (CDCl_3_): δ 1.51 (s, 9H), 5.91 (s, 1H), 7.45 (t, 1H, *J* = 7.2), 7.49–7.54 (m, 2H), 7.66 (dd, 2H, *J* = 7.2,1.2), 7.72–7.82(m, 4H); ^13^C NMR 600 (CDCl_3_): δ 28.5, 54.1, 114.2, 114.8, 119.8, 127.4, 127.4, 127.5, 128.6, 128.7, 129.2, 129.3, 130.4, 130.9, 132.7, 139.7, 144.4, 145.8, 151.6; *m*/*z* (EI): 353 (M+); *m*/*z* (EI): 353.1634 (M+, C_22_H_19_N_5_ calcd, 353.1635).

*(Z)-N-(2-Amino-1,2-diisocyanovinyl)-4-chlorobenzamide* **14c**. Brown crystals crystallized from EtOH (10%), mp 272 °C; ν_max_ (KBr)/cm^−1^: 3410, 3324, 3220, 3148, 2981, 2250, 2210, 1649, 1610, 1594, 1514, 1484, 1384, 1317, 1092, 1014, 846, 756; ^1^H NMR 400 (DMSO): δ 7.47 (s, 2H), 7.60 (d, 2H, *J* = 8.0 Hz), 7.95 (d, 2H, *J* = 8.0 Hz), 9.73 (s, 1H); ^13^C NMR 600 (DMSO): δ 88.9, 113.8, 116.9, 127.5, 128.4, 129.9, 131.7, 136.9, 164.2; *m*/*z* (EI): 246(M+); *m*/*z* (EI): 246.0302 (M+, C_11_H_7_ClN_4_O calcd, 246.0303).

*(Z)-N-(2-Amino-1,2-diisocyanovinyl)-4-methoxybenzamide* **14j**. Brown crystals crystallized from EtOH (15%), mp < 350 °C; ν_max_ (KBr)/cm^−1^: 3418, 3329, 3223, 2980, 2891, 2247, 2208, 1644, 1607, 1523, 1500, 1383, 1304, 1249, 1180, 1030, 844. ^1^H NMR 400 (CDCl_3_): δ 3.91 (s, 3H), 6.97 (d, 2H, *J* = 8.0 Hz), 8.08 (d, 2H, *J* = 8.4 Hz), 8.37 (s, 1H); *m*/*z* (EI): 242(M+); *m*/*z* (EI): 242.0799 (M+, C_12_H_10_N_4_O_2_ calcd, 242.0798).

*(Z)-N-(2-Amino-1,2-diisocyanovinyl)-4-phenylbenzamide* **14k**. Brown crystals crystallized from EtOH (45%), mp 251–252 °C; ν_max_ (KBr)/cm^−1^: 3414, 3325, 3220, 3063, 3038, 2992, 2250, 2211, 1651, 1641, 1608, 1523, 1503, 1484, 1384, 1320, 853, 747, 700; ^1^H NMR 400 (DMSO): δ 7.44 (m, 3H), 7.51 (t, 2H, *J* = 7.2 Hz), 7.76 (d, 2H, *J* = 7.6 Hz), 7.83 (d, 2H, *J* = 8.0 Hz), 8.02 (d, 2H, *J* = 8.4 Hz), 9.68 (s, 1H); ^13^C NMR 600 (DMSO): δ 89.4, 113.9, 117.0, 126.5, 126.9, 127.3, 128.2, 128.7, 129.0, 131.6, 138.9, 143.5, 164.9; *m*/*z* (EI): 288(M+); *m*/*z* (EI): 288.1003 (M+, C_17_H_12_N_4_O calcd, 288.1006). See Appendix A.

## 5. Conclusions

This study presents a facile new route leading to 5-(alkylamino)-6-aryl/alkylpyrazine-2,3-dicarbonitrile derivatives. The pyrazines are prepared starting from simple, readily available and inexpensive starting materials.

## Data Availability

Not applicable.

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
