# Peer review of "Construction of 5-(Alkylamino)-6-aryl/alkylpyrazine-2,3-dicarbonitriles via the One-Pot Reaction of Alkyl Isocyanides with Aryl/Alkyl Carbonyl Chlorides and Diaminomaleonitrile: Fluorescence and Antimicrobial Activity Evaluation"

_molecules, 2022, doi:10.3390/molecules27238278_

Round 1
Reviewer 1 Report
In the manuscript, 5-(alkylamino)-6-aryl/alkylpyrazine-2,3-dicarbonitriles were synthesized from alkyl isocyanides and aromatic and aliphatic acyl chlorides followed by reacition with aminomaleonitrile. The products were fully characterized, even by X-ray crystal structure analysis. The antimicrobial activity of the newly synthesized pyrazine-2,3-dicarbonitriles was assessed in addition to their UV and fluorescence results. It can be accepted for publication after a major revision.
Comments
1. The presentation of the manuscript should be reorganized and improved compeletely. There are two results and discussion sections now in the manuscript. They should be combined into one.
2. The proposed mechanism ir shown in very poor style, must be improved. SN2 reaction process cannot occur in C(sp2). All processes involving SN2(Csp2) should be addition-elimination process. Description on the mechanism should also be mentioned correctly.
3. Delete Scheme 3. its information is shown in Scheme 2.
4. Delete Table 1, move yields in scheme 2 (put them after the substituents). The mp data are already provided in experimental section.
5. all RC6H5 should be RC6H4.
6. English should be improved, even single/plural is incorrect.
Author Response
Dear Editor,
Thank you for your E-mail and for the presentational points made by the reviewers. Regarding our manuscript entitled: ‘Construction of 5-(alkylamino)-6-aryl/alkylpyrazine-2,3-dicarbonitrile via one pot reaction of alkyl isocyanides with aryl/alkyl carbonyl chlorides and diaminomaleonitrile: fluorescence and antimicrobial activity evaluation’ by Amal Al-Azmi and Elizabeth John submitted to Molecules.
The corrections according to the respected editor and reviewers’ comments are as following:
Reviewer #1 comments as follows:
In the manuscript, 5-(alkylamino)-6-aryl/alkylpyrazine-2,3-dicarbonitriles were synthesized from alkyl isocyanides and aromatic and aliphatic acyl chlorides followed by reaction with aminomaleonitrile. The products were fully characterized, even by X-ray crystal structure analysis. The antimicrobial activity of the newly synthesized pyrazine-2,3-dicarbonitriles was assessed in addition to their UV and fluorescence results. It can be accepted for publication after a major revision.
Comments
- The presentation of the manuscript should be reorganized and improved completely. There are two results and discussion sections now in the manuscript. They should be combined into one.
Reply: The two results and discussion sections are now combined into one.
- The proposed mechanism is shown in very poor style, must be improved. SN2 reaction process cannot occur in C(sp2). All processes involving SN2(Csp2) should be addition-elimination process. Description on the mechanism should also be mentioned correctly.
Reply: We would like to thank the referee for this comment. The mechanism has been altered.
- Delete Scheme 3. its information is shown in Scheme 2.
Reply: Scheme 3 is now deleted as suggested.
- Delete Table 1, move yields in scheme 2 (put them after the substituents). The mp data are already provided in experimental section.
Reply: We reached to a compromised solution, as referee 1 suggested to delete table 1 and to add yields in scheme 2, while referee 3 suggested to keep it. Table 1 is kept with adding the yields and substituents were removed from scheme 2 and are added in table 1.
- All RC6H5 should be RC6H4.
Reply: Required corrections are done.
- English should be improved, even single/plural is incorrect.
Reply: English language is revised and improved.
Reviewer 2 Report
The creation of new molecules and the addition of novel scaffolds to molecular libraries are always have been fascinating work. The utility of these molecules makes them socio-impactful. The present manuscript describes the one-pot synthesis of pyrazine-2,3-dicarbonitrile from the reaction of alkyl isocyanides with aryl/alkyl carbonyl chlorides and diamino-malononitrile. In addition, the prepared molecules' fluorescence and antimicrobial activity were evaluated. The experimental and other studies are performed well and described well. However, the parts of this work are not entirely new. The overall idea is interesting, and I can therefore recommend this work for publication in the molecules after revision and incorporation of the following suggestions:
1. In the introduction section, detailed description of Scheme 1 (see page 3) is unnecessary and should be removed.
2. The authors should include the fluorescence spectra of all the compounds.
3. The results of the antimicrobial evaluation are not very promising, and some of the compounds are ineffective. The authors should determine the MIC for the tested compounds.
4. The fluorescence results of a few compounds seem very interesting; therefore, if possible, the authors should explore the application profile of these compounds based on their fluorescence.
5. Some language and general representation errors should be rectified. e.g., standard abbreviation for milliliter, 'mL' should be used instead of 'ml'.
Author Response
Dear Editor,
Thank you for your E-mail and for the presentational points made by the reviewers. Regarding our manuscript entitled: ‘Construction of 5-(alkylamino)-6-aryl/alkylpyrazine-2,3-dicarbonitrile via one pot reaction of alkyl isocyanides with aryl/alkyl carbonyl chlorides and diaminomaleonitrile: fluorescence and antimicrobial activity evaluation’ by Amal Al-Azmi and Elizabeth John submitted to Molecules.
The corrections according to the respected editor and reviewers’ comments are as following:
Reviewer #2 comments as follows:
Comments and Suggestions for Authors
The creation of new molecules and the addition of novel scaffolds to molecular libraries are always have been fascinating work. The utility of these molecules makes them socio impactful. The present manuscript describes the one-pot synthesis of pyrazine-2,3-dicarbonitrile from the reaction of alkyl isocyanides with aryl/alkyl carbonyl chlorides and diamino-malononitrile. In addition, the prepared molecules' fluorescence and antimicrobial activity were evaluated. The experimental and other studies are performed well and described well. However, the parts of this work are not entirely new. The overall idea is interesting, and I can therefore recommend this work for publication in the molecules after revision and incorporation of the following suggestions:
- In the introduction section, detailed description of Scheme 1 (see page 3) is unnecessary and should be removed.
Reply: Description of Scheme is removed.
- The authors should include the fluorescence spectra of all the compounds.
Reply: The fluorescence spectra of all the compounds are now included.
- The results of the antimicrobial evaluation are not very promising, and some of the compounds are ineffective. The authors should determine the MIC for the tested compounds.
Reply: We thank the referee for his/her comment. This study explores the outcome from the reactions of DAMN, carbonyl chlorides and isocyanides. The antimicrobial activity investigation is carried out as a preliminary study to explore their reactivity as potential antimicrobial reagents and to determine MIC as suggested above is not in the scope of this study. However, we would have wanted to do MIC but unfortunately, the biologist who carried out the study has resigned 6 weeks ago, and there is no other biologist available. The MIC determination will be considered in future work.
- The fluorescence results of a few compounds seem very interesting; therefore, if possible, the authors should explore the application profile of these compounds based on their fluorescence.
Reply: We thank the referee for his/her positive comments concerning the applications of these compounds. We are in the process to start new research to carry OLED applications on the prepared compounds and on other new derivatives.
- Some language and general representation errors should be rectified. e.g., standard abbreviation for milliliter, 'mL' should be used instead of 'ml'.
Reply: Correction is done.
Reviewer 3 Report
A. Al-Azmi and E. John submitted a manuscript intitled: Construction of 5-(alkylamino)-6-aryl/alkylpyrazine-2,3-dicarbonitrile via one pot reaction of alkyl isocyanides with aryl/alkyl carbonyl chlorides and diaminomaleonitrile: fluorescence and antimicrobial activity evaluation.
Pyrazines are well-known nitrogen containing heterocyclic compound with two nitrogen atoms in 1,4-positions of the six membered ring. They are considered one of the most significant classes of nitrogen heterocyclic compounds which possessing numerous applications well explained within the manuscript. Pyrazines in very interesting molecules and their synthesis is a great challenge for chemists. For these reasons the argument is very actually and researched.
Manuscript is good structured in all part but, before your publication, some minor changes are needed. Synthesis is very simple and the molecules are very well characterized. The modifications to be made are reported in the attached file and once the manuscript has been changed, it can be accepted for its publication.
Best Regards.

Author Response
Dear Editor,
Thank you for your E-mail and for the presentational points made by the reviewers. Regarding our manuscript entitled: ‘Construction of 5-(alkylamino)-6-aryl/alkylpyrazine-2,3-dicarbonitrile via one pot reaction of alkyl isocyanides with aryl/alkyl carbonyl chlorides and diaminomaleonitrile: fluorescence and antimicrobial activity evaluation’ by Amal Al-Azmi and Elizabeth John submitted to Molecules.
The corrections according to the respected editor and reviewers’ comments are as following:
Reviewer #3 comments as follows:
Comments and Suggestions for Authors
- Al-Azmi and E. John submitted a manuscript intitled: Construction of 5-(alkylamino)-6-aryl/alkylpyrazine-2,3-dicarbonitrile via one pot reaction of alkyl isocyanides with aryl/alkyl carbonyl chlorides and diaminomaleonitrile: fluorescence and antimicrobial activity evaluation.
Pyrazines are well-known nitrogen containing heterocyclic compound with two nitrogen atoms in 1,4-positions of the six membered ring. They are considered one of the most significant classes of nitrogen heterocyclic compounds which possessing numerous applications well explained within the manuscript. Pyrazines in very interesting molecules and their synthesis is a great challenge for chemists. For these reasons the argument is very actually and researched.
Manuscript is good structured in all part but, before your publication, some minor changes are needed. Synthesis is very simple, and the molecules are very well characterized. The modifications to be made are reported in the attached file and once the manuscript has been changed, it can be accepted for its publication.
Best Regards.
Reply: All comments and corrections made by referee 3 in the pdf file (peer-review-23940473.v2.pdf) have been taken into consideration and corrected and/or added.
Round 2
Reviewer 1 Report
The drawing on the mechanism can be further improved for more professional presentation.